# Quantitative Textural and Rheological Data on Different Levels of Texture-Modified Food and Thickened Liquids Classified Using the International Dysphagia Diet Standardisation Initiative (IDDSI) Guideline

**DOI:** 10.3390/foods12203765

**Published:** 2023-10-13

**Authors:** Man Chun Wong, Karen M. K. Chan, Tsz Ting Wong, Ho Wah Tang, Hau Yin Chung, Hoi Shan Kwan

**Affiliations:** 1Food Research Centre, School of Life Sciences, The Chinese University of Hong Kong, Hong Kong, Chinaanthonychung@cuhk.edu.hk (H.Y.C.); 2Swallowing Research Laboratory, The University of Hong Kong, Hong Kong, China; kenxes@hku.hk

**Keywords:** dysphagia, International Dysphagia Diet Standardisation Initiative (IDDSI), texture profile analysis, hardness, viscosity, thickener

## Abstract

Diet modification is a common compensation strategy to promote swallowing safety in patients with swallowing difficulties. The International Dysphagia Diet Standardisation Initiative (IDDSI) guideline provides qualitative descriptions on texture-modified food and thickened liquid. This study aimed to establish quantitative textural and rheological data on different IDDSI levels based on common Chinese ingredients and dishes. Textural and rheological properties of 226 samples of various food textures and 93 samples of various liquid consistencies were obtained using a texture profile analysis (TPA) and viscometer, respectively. The establishment of such quantitative data can be used for future texture-modified food product development and research purposes.

## 1. Introduction

### 1.1. Dysphagia

Swallowing refers to the act of eating and drinking. In the process of normal swallowing, solid food is placed inside the oral cavity for mastication, which breaks the food into lumps and further into a bolus that is mixed with saliva. When the bolus is propelled into the oropharynx, the airway is temporarily closed in order to prevent the bolus from entering the trachea. The bolus then enters the oesophagus, which transfers the bolus to the stomach for digestion. The process of swallowing liquid is similar, except that mastication usually is not necessary [1]. Failure in transferring food or liquid from the oral cavity to the stomach safely is termed as dysphagia [2]. Patients with neurological disorders such as stroke, dementia, and Parkinson’s disease are at a high risk of having dysphagia [3]. Common symptoms of dysphagia include leakage of food or liquid from the lips, prolonged mastication, choking, and coughing before and after swallowing [4]. If dysphagia management is not implemented properly, patients may experience malnutrition, dehydration, aspiration, or even asphyxiation, which could be fatal [5].

### 1.2. Diet Modification

Diet modification is defined as the alternation of food texture and liquid consistency [4]. It serves as a common compensatory strategy for patients with dysphagia to promote swallowing safety [2]. Speech therapists prescribe different levels of texture-modified foods and liquids to people with dysphagia based on their individual needs and swallowing abilities. The ultimate aims of diet modification are to find consistencies that the person can safely take orally, with minimal risk of aspiration, and be as liberal as possible to maintain quality of life and independence. Solid food modification often involves changing the “degree of structure” of the food, including its hardness, adhesiveness, and cohesiveness [6]. As the food texture changes, the physiology involved in swallowing the food also changes. For example, with softer food, the number of chewing cycles, the number of squeezing actions of the tongue, and the tongue force required for crushing and compressing the food decrease [7,8]. A study also found that less saliva is produced when the food texture becomes smoother and moister [8].

Thickened liquid is prescribed to reduce the risk of penetration and aspiration [9]. As the consistency of a liquid increases, the shear viscosity and the extensional viscosity also increase accordingly. Shear viscosity refers to the speed of flow, or thickness, of the liquid [6]. Higher shear viscosity results in longer pharyngeal transit time, in which the liquid flows more slowly in the oropharynx, allowing more time for airway closure. Extensional viscosity refers to the internal bonding strength, or cohesiveness, of the liquid [6]. Higher extensional viscosity results in shorter liquid elongation, in which the liquid forms a more cohesive bolus, preventing liquid droplets from being fragmented and causing post-swallow residue at the oropharynx. Through swallowing assessment, a speech therapist finds the appropriate thickness level at which the patient has enough strength to swallow and to swallow without aspiration and without too much residue remaining in the oropharyngeal tract.

### 1.3. International Dysphagia Diet Standardisation Initiative

Before the publication of the International Dysphagia Diet Standardisation Initiative (IDDSI) guideline in 2015, there was no standardised terminology or definition being used globally to describe different types of a modified diet [10]. The size, texture, and consistency varied from setting to setting, which, in turn, put a patient’s swallowing safety into jeopardy. The alignment in terminology and definition proposed by the IDDSI guideline enables patients, caregivers, clinicians, and industrial and frontline workers to have a mutual understanding towards the targeted dysphagia diet. The guideline consists of 9 hierarchical levels, with liquid being classified into levels 0 to 4 and food being classified into levels 3 to 7. Level 0 refers to thin liquid (e.g., water); as the level increases, the thickness of the liquid increases, with level 4 representing extremely thick liquid. Levels 3 and 4 can also be used to describe liquidised and pureed food, respectively. As the food level increases, the food particle size increases and less processing is needed. Level 5 represents minced and moist; level 6, soft and bite-size; level 7EC, as easy to chew; and level 7 represents regular food. Standardised definitions and descriptions of the different levels of a dysphagia diet are included in the guideline. Testing methods are available at each level for users to determine if the food or liquid matches the proposed requirement using handy equipment. For example, a 10 mL syringe can be used to conduct a flow test to check the liquid consistency; a fork can be used to measure the size of a food lump using the slots between the fork prongs.

### 1.4. Purpose of the Current Study

Despite the simplicity and user-friendliness of the IDDSI guideline, descriptions regarding the food texture and liquid consistency are relatively qualitative in nature. No quantitative data on the textural and rheological properties of food and liquid are available. The absence of such data poses difficulty for industrial partners to assure the quality of their modified diet products [11]. Previous research consistently indicated that discrepancy existed between a manufacturer’s instruction and the IDDSI guideline in terms of liquid consistency [12,13]. The thickened liquids prepared according to a manufacturer’s instruction were either too thin or too thick to be consumed by patients with dysphagia. The absence of quantitative data, additionally, limits researchers in conducting studies investigating the impact of food texture and liquid rheology on swallowing [6]. Using only qualitative measurements fails to reveal subtle differences for food or liquid under the same IDDSI level [14]. As a result, this study aimed to establish quantitative data on the textural and rheological properties of food or liquid that can be used to accompany and supplement the IDDSI guideline.

## 2. Materials and Methods

### 2.1. Sample Preparation and IDDSI Level Determination

Sixty food ingredients and dishes that are popular in Hong Kong or Asian communities were chosen for this study (Appendix A). A total of 226 food samples with different textures were prepared from the food ingredients and dishes by cutting them into different sizes and cooking using conventional methods, such as steaming, boiling, and frying for different durations. For samples at IDDSI level 4, the food was pureed by blending with water or further remoulded into shaped meals using Suberakaze Enzyme Gellant (Foodcare, Sagamihara, Japan). The IDDSI levels of the food samples were determined by a fork pressure test, spoon tilt test, and fork drip test according to IDDSI guidelines [10]. A total of 93 liquid samples of drinks and condiments of different thicknesses were prepared from 15 of the food ingredients and six dishes (Appendix A). The samples were thickened by either a xanthan gum-based thickener (Thick & EasyTM Clear, Fresenius Kabi, Austin, TX, USA) or a commercial corn starch following common cooking practices. The IDDSI flow test was administered according to the IDDSI guidelines [10] to determine the level of thickened liquid food. Specifically, a BD 10 mL syringe (REF 302143) with a measured length of 61.5 mm from the 0 mL line to the 10 mL line was used to measure the volume of the sample remaining from 10 mL after 10 s of flow. Level 4 and Level 3 samples were further differentiated with the IDDSI fork drip test. Appendix A summarize the cooking and preparation methods of the samples.

### 2.2. Instrumentation and Measurement Procedures

#### 2.2.1. Texture Profile Analysis

Hardness, adhesiveness, and cohesiveness of the samples were determined by a texture profile analysis (TPA) using a TA.XTplus Texture Analyser (Stable Micro System, Godalming, UK). The testing protocol was adopted from the Universal Design Food (UDF) Test established by the Japan Care Food Conference (2016) [15]. Each sample was transferred to a container of 40 mm in diameter. The sample was filled up to a 15 mm height. Using a 20 mm diameter cylinder probe (P/20), the sample was compressed twice at a speed of 10 mm/sec and a clearance of 5 mm. The force during the compression was recorded by the texture analyser to generate the texture analysis profile. The texture profile parameters were determined as follows: (1) hardness was defined as the maximum force required for compressing foods and was calculated as the peak force of the first compression; (2) adhesiveness was calculated as the negative area for the first bite, representing the work necessary to pull the compressing plunger away from the sample; and (3) cohesiveness was the ratio of the integrated energy required for the second compression to that of the first compression [16,17]. Four replicates were prepared and measured for each sample.

#### 2.2.2. Rheology Analysis

The viscosity of the liquid samples was measured with a Brookfield DV2TRV Viscometer (Brookfield Engineering Laboratories, Inc., Middleboro, MA, USA) at a shear rate of 50 s^−1^, which simulated the shear rate during swallowing [18,19,20]. A UL adapter or a small sample adaptor (SSA) with spindle SC4-21 or SC4-28 was used for measurement depending on the viscosity range of each sample and % torque. Table 1 details the viscosity range of each spindle at a 50 s^−1^ shear rate. According to the manufacturer’s operating instructions, viscosity measurements should be accepted within the equivalent % torque range from 10% to 100% for any combination of spindle/speed rotation. The viscosity was measured in centipoise (cP) at room temperature and for 10 min. The average viscosity at the last minute was re-ported. There were three technical replicates for each sample, and each replicate was measured twice. Therefore, six measurements were obtained for each sample.

#### 2.2.3. Statistical Analysis

A Fisher’s exact test was run to analyse the association between cohesiveness and adhesiveness to the perception of the stickiness of the food items. A simple linear regression was carried out to analyse the independence of viscosity on the flow test remaining volume, and the significance was set at *p* ≤ 0.05. A regression equation and regression square were obtained from the correlation of the remaining volume of the flow test and the natural logarithm of the viscosity of the different ingredients at different IDDSI levels.

## 3. Results

### 3.1. Texture Profile Analysis

The texture profile analysis showed us the characteristics of the food. Detailed values of hardness, adhesiveness, and cohesiveness of the food items are shown in Appendix A. Table 2 summarizes the ranges of the three parameters of the different types of food items.

According to the IDDSI spoon tilt test, seven samples were classified as too sticky, as a great deal of food residue was left on the spoon while the spoon was tilted. The spoon tilt test is recommended by the IDDSI to test the stickiness of food samples in levels 4 and 5, as thicker food may become too sticky, which poses an aspiration risk. These failed samples were purees of potato, rice, pumpkin, taro, chestnut, black fungus, and snow fungus. In this study, we found that both adhesiveness and cohesiveness were associated with the perception of the stickiness of food items (Figure 1). Statistical analysis indicated that there was a significant association between cohesiveness and adhesiveness to stickiness; both *p*-values were less than 0.0001. In general, most samples of high perceptual stickiness (lots of residue remaining on a spoon) had high cohesiveness of over 0.8 and adhesiveness lower than −25 g·s. Some sticky samples had cohesiveness of over 0.65 and adhesiveness of lower than −55 g·s (Figure 1).

### 3.2. Rheology Analysis

The viscosity of the triplicate samples was measured twice, and their mean viscosities corresponding to the remaining volume of the syringe flow test are reported (Appendix A). The samples were categorized into groups based on the thickening agent (Table 3). Figure 2 shows the distribution of all the samples based on their viscosity and the result of the IDDSI flow test. The samples were categorized into five IDDSI levels. As the IDDSI level increased according to the IDDSI flow test, there was a corresponding increase in viscosity. An exponential relationship was observed.

As the viscosity of level 0 and level 4 could be extremely low and high, which was beyond the test limit of the syringe flow test, only samples of level 1 to level 3 were used to study the correlation between viscosity and the syringe flow test (Appendix A). A significant regression equation was obtained (y = 0.269x + 3.3619; R^2^ = 0.92), correlating the natural logarithm of viscosity with the remaining volume of the flow test. It indicated that the two parameters were positively correlated. Viscosity is a reliable predictor of the IDDSI level of the samples.

## 4. Discussion

### 4.1. Texture Profile Analysis

#### 4.1.1. Comparison of Hardness to International Standard

The hardness value of the food samples describes the force required to compress the food during chewing. Foods with higher hardness are harder for consumers to chew and break down. The Japan Care Food Conference established the Universal Design Foods (UDF) concept, which classifies food into four categories according to hardness [21,22]. In 2019, the Food Industry Research and Development Institute (FIRDI) in Taiwan launched the “Eatender” labelling system for elder-friendly foods. FIRDI set up the food classification standards for four texture specifications, ranging from “easy-to-chew”, “gum-chewable”, and “tongue crushing” to “no chewing” [23]. The categories of texture-modified food in different schemes were matched, and standards of hardness are listed in Table 4.

In this study, food samples were prepared to different IDDSI levels; the hardness of the samples of Levels 7 to 4 were lower than 500 × 10^3^ N/m^2^, 50 × 10^3^ N/m^2^, 20 × 10^3^ N/m^2^, and 5 × 10^3^ N/m^2^, respectively. This aligned with the hardness standard of the UDF and Eatender, except for level 4 dried bonito (8.84 × 10^3^ N/m^2^). The IDDSI guideline recommends that food sizes for level 6 should be smaller than 1.5 cm × 1.5 cm × 1.5 cm and there was no size restriction for level 7. In the current study, in order for the meat samples, such as beef, chicken, and pork, to pass the IDDSI fork pressure test and meet the hardness range of the UDF, the meat sizes had to be further reduced as compared with the IDDSI standard. Hard vegetables with plenty of fibre, like choy sum, pumpkin, taro, and white radish, also had to be cut into pieces smaller than 0.75 cm × 1.5 cm × 1.5 cm or even 0.75 cm × 0.75.cm × 1.5 cm to make them soft enough to meet the IDDSI testing methods. Hard fruits such as apples and pineapples could either be cut into smaller sizes or cooked to reduce the hardness. For dishes, most of the level 4 dishes in a shaped-meal form were slightly harder than those in a puree form. As the shaped meals were remoulded into shape, their hardness would be increased by the gelatinisation of the gelling agent in order to maintain their shapes.

#### 4.1.2. Adhesiveness and Cohesiveness

Adhesiveness and cohesiveness are two common parameters that are investigated when using a texture profile analysis [25,26,27]. For adhesiveness, the more negative the value, the more the food adheres to the surface of other materials. If the adhesiveness is 0, it means that the food samples do not stick to the testing probe. There is a general trend of higher adhesiveness at level 7 and level 4 and lower adhesiveness at level 5 and level 6. Moisture content, food ingredient, and particle size could be possible factors that influenced the adhesiveness [28]. Level 7 samples had larger surface areas, so they tended to stick to the testing probe. Level 4 samples were blended into purees and contained more moisture; as a result, they were more adhesive.

For cohesiveness, if the value is closer to 1, it means that it can maintain its structure during chewing. Although the cohesiveness of different food samples was quite diverse at level 7, the range of values became more homogeneous as samples moved down IDDSI levels. The range decreased from 0.08–0.83 for level 7 to 0.53–0.89 for level 4. The cohesiveness was inversely related to the particle size [29,30]; so, the cohesiveness of most of the food samples was the highest when they were pureed. Having higher adhesiveness and cohesiveness, pureed food was stickier than diced or minced food. We should pay special attention to pureed foods to ensure that they are not too sticky to swallow.

In the current study, food samples that had adhesiveness over −25 g·s and cohesiveness higher than 0.8 were too sticky to pass the IDDSI spoon tilt test. For example, level 4 potato (adhesiveness: −73.741 g·s and cohesiveness: 0.840) was too sticky after blending with water. There was a great deal of residue on the spoon after performing the spoon tilt test. Another example was rice. Rice became stickier and stickier when it was adjusted from level 7 to level 4. The adhesiveness and cohesiveness of the level 4 rice sample were −39.924 g·s and 0.816, respectively. It was extremely sticky and glued to the spoon during the spoon tilt test. Other starchy food items, like taro, pumpkin, and chestnut, as well as black fungus and snow fungus, which are rich in polysaccharides, also encountered this problem. They are not recommended to be eaten individually. A prior study showed that the stickiness of food was highly correlated to the ratio of amylose and amylopectin [31]. To reduce the stickiness of such sticky foods, they should be mixed with other ingredients to lower the cohesive force among food particles or blended with more water; however, thickener will be needed to maintain their thickness at level 4 if the puree becomes too watery.

#### 4.1.3. Concerning Descriptive Properties

One of the major differences between the IDDSI and other international guidelines is its descriptive characteristics. Apart from hardness and stickiness, the guideline also specifies the particle size and chewiness of the food items. To meet IDDSI level 4, food samples should be smooth with no lumps and minimal granulation. However, the pureed food particles of level 4 dried shrimp (hardness: 3.936 × 10^3^ N/m^2^) and dried bonito (hardness: 8.835 × 10^3^ N/m^2^) tended to stick together and form lumps. Therefore, these ingredients are not recommended to be consumed individually. They should be blended with other ingredients to prevent the food particles from forming lumps.

The hardness of some samples was within the hardness range, but they were too tough and chewy to be broken apart easily by the side of a fork or a spoon. Dried octopus, even when it was cut into smaller sizes to meet the hardness standard (Level 7: 20.462 × 10^3^ N/m^2^ and level 6: 15.275 × 10^3^ N/m^2^), could not pass the fork pressure test. For shiitake mushroom, its hardness (level 7: 135.528 × 10^3^ N/m^2^ and level 6: 34.425 × 10^3^ N/m^2^) was also within the hardness range, but it was too tough to be cut by the side of a fork or a spoon. Therefore, it did not fulfil the requirement of IDDSI level 7EC and 6. In fact, there are more food items in Chinese cuisine that are chewy in texture, particularly preserved vegetables and dried seafood. Therefore, other than obtaining the hardness of the food, the IDDSI testing method is also recommended to ensure the food satisfies other subjective properties.

### 4.2. Rheology Analysis

There are two main international guidelines on viscosity ranges in the world, and they are the National Dysphagia Diet (NDD) [18] and the Japanese Dysphagia Diet 2013 by the Japanese Society of Dysphagia Rehabilitation (JSDR) [32]. However, there is a significant difference between the viscosity range in the NDD and that in the JSDR even though they were measuring at the same shear rate of 50 s^−1^. Some research suggested that it was due to the prevalence of the different types of thickening agents in different countries [32]. A starch-based thickener was common in Western countries during the establishment of the NDD, while a gum-based thickener has been more popular in Japan. The ranges of viscosity of the samples thickened by a gum-based thickener and corn starch are shown in Table 5 and compared to the international standard. Among the 14 samples thickened by Thick & EasyTM Clear, the viscosity values were similar to the range provided by the JSDR. For the food items thickened by corn starch, their viscosities were more similar to the range of the NDD.

Among all the liquid samples tested, scallion oil and peanut sauce behaved quite differently from the other samples (Figure 2). Scallion oil was determined as level 2 using the IDDSI methods, but its viscosity (57.6 cP) was much lower than other level 2 samples. Peanut sauce was determined as level 3 by the IDDSI flow test, but its viscosity (882.0 cP) was much higher than the distribution of other level 3 samples. The discrepancy may be due to the high oil content of both samples, while the other samples mainly consisted of water. The rheological property of oil is quite different from water. Oil has a lubricating property, but it has higher resistance to flow [33]. It can be sheared easily but drops slowly. Therefore, liquid with a high oil content did not behave in the same way as the water-based liquid and did not follow the distribution of the other samples.

Research showed that viscosity can be affected by pH, temperature, fat content, density, the medium being thickened, and the thickener itself [34,35,36]. Meat soup has a high fat content, vegetable soup is rich in polysaccharides or fibre, and fruit juice contains lots of sugar. These affected the ability of the thickener to thicken the liquid. It may be inappropriate to compare each food item directly with the viscosity values reported in the past since the details of the medium and thickening agent were not provided. Different food items have their own physical properties so their viscosity range should be different from each other. Further studies are needed to investigate the viscosity range of each food.

## 5. Conclusions

The current study reported the hardness, adhesiveness, and cohesiveness values of food samples and viscosity values of liquid samples at different IDDSI levels. All samples were tested with the IDDSI testing methods to ensure they matched the IDDSI level requirements. The selected samples were mainly based on Chinese dishes, and some Chinese ingredients may need to be avoided or cut into smaller-than-IDDSI-specified sizes to meet the IDDSI-required softness. The majority of the samples aligned with international hardness standards, while sticky samples had higher cohesiveness and adhesiveness values. Rheology analysis explored the viscosity of the food samples, and a positive correlation was observed between viscosity and IDDSI flow test results, establishing viscosity as a reliable predictor for assessing the IDDSI level of the samples.

The results provide a reference guide for preparation, development, and further food testing of texture-modified foods that accurately meet the needs of individuals with swallowing difficulties. Further studies are recommended to explore the characteristics and properties of specific food items in more detail, taking into account the influence of factors such as pH, temperature, and the composition of the medium being thickened. These findings can contribute to a better understanding of the physical properties of different food items and the development of more precise guidelines for texture-modified foods.

## Figures and Tables

**Figure 1 foods-12-03765-f001:**
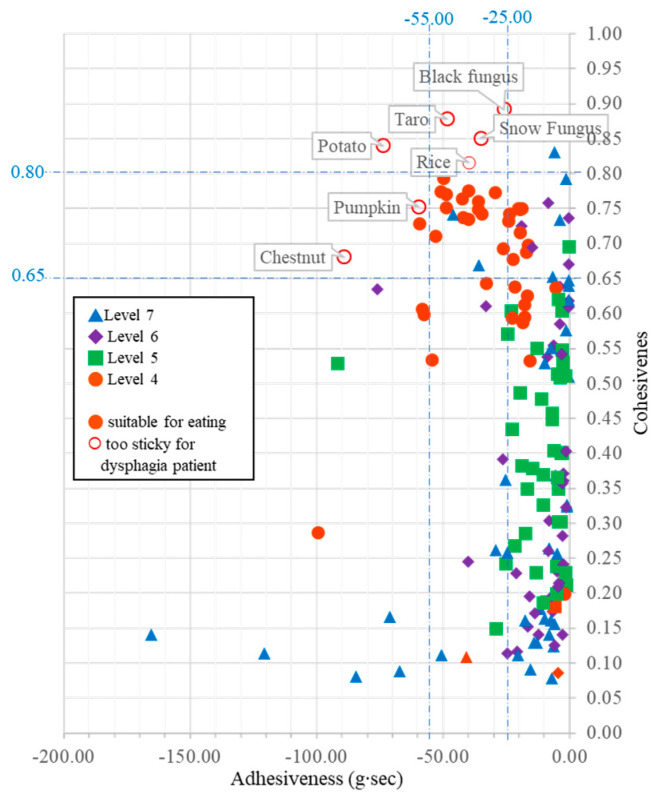
Scatter plot of mean adhesiveness and mean cohesiveness of the food samples measured by TPA method. Seven samples that were defined as too sticky by IDDSI spoon tilt test are denoted with their sample name. The dotted lines denote the estimated thresholds for high, medium, and low cohesiveness and adhesiveness.

**Figure 2 foods-12-03765-f002:**
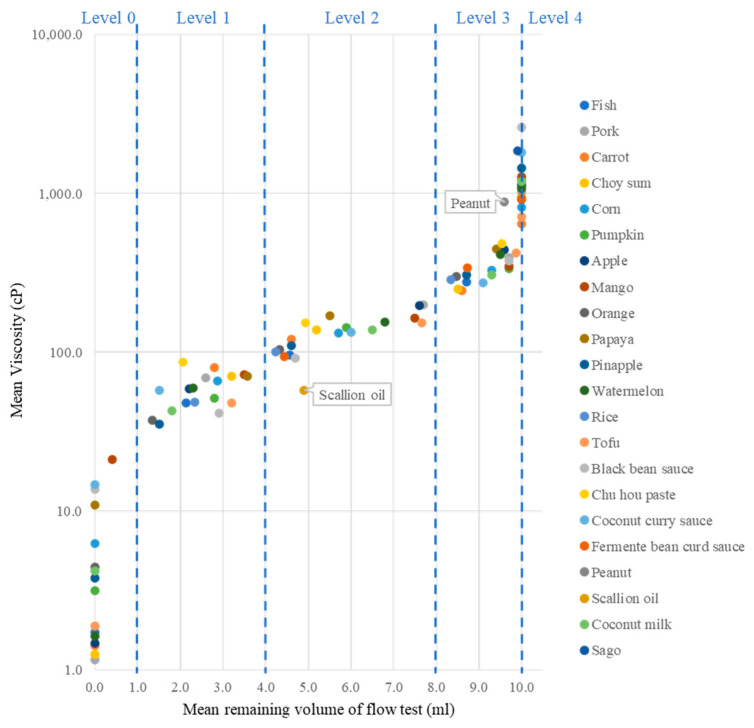
Scatter plot of mean remaining volume of flow test and mean viscosity of the liquid samples measured (*n* = 6).

**Table 1 foods-12-03765-t001:** Specifications of different spindles used in this study.

Spindle	Sample Volume (mL)	Shear Rate Constant (SRC)	RPM at 50 s^−1^ Shear Rate	Full Scale Viscosity Range (cP) at 50 s^−1^ Shear Rate
UL	16	1.223	40.9	15.6–156.544
SC4-21	7.1	0.93	53.8	93–930
SC4-28	11.0	0.28	178.6	280–2800

**Table 2 foods-12-03765-t002:** The ranges of hardness, adhesiveness, and cohesiveness of different types of food.

Type	IDDSI Level	Hardness (×10^3^ N/m^2^)	Adhesiveness (g·s)	Cohesiveness
Ingredients				
Fish and meat	7	53.97–131.76	(−25.06)–0.00	0.36–0.73
	6	14.49–48.52	(−8.03)–0.00	0.55–0.76
	5	8.22–12.53	(−4.61)–0.00	0.51–0.70
	4	0.85–3.52	(−58.19)–(−22.63)	0.53–0.74
Vegetables	7	11.00–392.50	(−120.48)–0.00	0.08–0.83
	6	1.97–49.49	(−26.18)–(−1.13)	0.09–0.72
	5	1.66–17.08	(−22.66)–(−1.16)	0.18–0.55
	4	0.51–2.19	(−50.69)–(−16.13)	0.64–0.89
Fruits	7	40.27–347.15	(−84.40)–(−7.31)	0.08–0.26
	6	17.84–47.12	(−20.66)–(−2.35)	0.12–0.30
	5	1.56–13.92	(−19.58)–(−1.61)	0.23–0.49
	4	0.45–4.84	(−42.20)–(−17.70)	0.59–0.78
High-protein food	7	5.18–370.10	(−32.75)–0.00	0.55–0.79
	6	3.70–46.98	(−23.16)–0.00	0.54–0.74
	5	0.89–9.05	(−24.42)–(−3.06)	0.38–0.63
	4	0.67–3.94	(−53.03)–(−5.51)	0.52–0.74
High-starch food	7	5.09–287.07	(−165.12)–(−0.94)	0.09–0.67
	6	3.14–45.05	(−75.95)–(−0.94)	0.11–0.63
	5	3.75–18.85	(−91.72)–(−3.23)	0.15–0.53
	4	0.61–3.03	(−89.16)–(−18.59)	0.68–0.88
Dishes				
Fish	7	34.84–303.80	(−62.14)–0.00	0.41–0.75
	6	8.66–47.76	(−24.91)–(−3.14)	0.40–0.66
	5	7.30–16.01	(−33.29)–(−14.57)	0.30–0.46
	4 (soft meal)	3.05–4.91	(−20.16)–(−11.38)	0.43–0.57
	4 (puree)	0.72–2.44	(−66.40)–(−20.12)	0.62–0.74
Meat	7	85.68–151.07	(−223.65)–(−4.31)	0.08–0.62
	6	9.41–25.98	(−23.60)–(−5.92)	0.48–0.59
	5	1.32–18.94	(−26.00)–(−12.57)	0.36–0.58
	4 (soft meal)	2.16–3.71	(−14.60)–(−9.11)	0.41–0.89
	4 (puree)	0.93–4.55	(−39.16)–(−11.14)	0.49–0.99
Condiments	7	13.56–108.19	(−29.94)–(−2.29)	0.28–0.49
	6	13.56–26.27	(−29.94)–(−2.29)	0.30–0.49
	5	6.75–13.84	(−17.66)–(−15.26)	0.26–0.41
	4 (soft meal)	--	--	--
	4 (puree)	0.92–3.18	(−49.71)–(−33.97)	0.70–0.80
Dessert	7	17.01–115.15	(−47.16)–(−15.81)	0.14–0.80
	6	--	--	--
	5	0.89	(−11.68)	0.65
	4 (soft meal)	1.44–4.61	(−14.62)–(−9.93)	0.50–0.62
	4 (puree)	--	--	--
Others	7	27.22–114.07	(−61.95)–(−20.66)	0.57–0.64
	6	9.05–30.29	(−58.11)–(−2.70)	0.52–0.84
	5	10.75–18.39	(−9.10)–(−2.23)	0.52–0.62
	4 (soft meal)	1.75–4.79	(−20.34)–(−12.35)	0.36–0.49
	4 (puree)	3.88–4.84	(−31.33)–(−7.83)	0.47–0.50

**Table 3 foods-12-03765-t003:** The mean viscosity of the food items thickened by different thickening agents.

	Mean Viscosity (cP)				
IDDSI Level	4	3	2	1	0
Thickened by Thick & Easy^TM^ Clear					
Fish soup	1101.3	275.6	95.8	47.8	1.7
Pork soup	1032.5	395.0	198.1	69.4	1.2
Carrot juice	645.5	244.0	120.2	79.9	1.4
Choy sum soup	1085.7	251.1	137.6	70.7	1.3
Corn soup	813.9	327.4	132.4	66.1	6.3
Pumpkin soup	1225.3	334.5	143.2	51.4	3.1
Apple juice	1132.0	442.1	197.8	58.8	1.5
Mango juice	1271.3	349.4	164.3	72.6	21.2
Orange juice	918.4	298.8	104.3	37.3	4.4
Papaya juice	945.5	446.7	169.3	70.6	10.9
Pineapple juice	1440.0	306.6	110.3	35.4	3.8
Watermelon juice	1071.7	413.3	154.7	59.9	1.6
Soybean milk	713.1	423.2	152.5	48.2	1.9
Coconut milk	1192.0	307.5	138.2	43.0	4.2
Thickened by corn starch					
Black bean sauce	2625.3	377.6	91.7	41.6	13.6
Chu hou paste	938.9	483.3	153.8	86.9	
Coconut curry sauce	1810.0	274.3	133.9	57.8	14.6
Red fermented bean curd sauce	929.6	340.4	94.2		
No thickener added					
Congee (rice)		287.7	101.0	48.2	
Peanut sauce		882.0			
Scallion oil			57.6		

**Table 4 foods-12-03765-t004:** International standard for the hardness of texture-modified food.

Scheme/Country		<Regular Food			Extensively Texture-Modified Food>
Universal Design Foods (UDF), Japan (Japan Care Food) [21,22]	Categories	Stage 1 Able to chew easily	Stage 2 Able to smash with gum	Stage 3 Able to smash with tongue	Stage 4 Able to swallow without chewing
	Hardness (N/m^2^)	<5 × 10^5^	<5 × 10^4^	<2 × 10^4^	<5 × 10^3^
Eatender, Taiwan [23,24]	Categories	Easy-to-chew	Gum-chewable	Tongue crushing	No chewing
	Hardness (N/m^2^)	<5 × 10^5^	<5 × 10^4^	<2 × 10^4^	<5 × 10^3^
IDDSI [10] *	Categories	Level 7 Easy to chew	Level 6 Soft and bite size	Level 5 Minced and moist	Level 4 Pureed
	Hardness assessed by fork pressure test	Can be broken down with pressure from fork; thumbnail blanches to white	Can be broken down with pressure from fork; thumbnail blanches to white	Particles easily come through the tines of a fork; thumbnail does not blanch to white	Tines of a fork make a clear pattern on the surface
Current study	Hardness of tested food items (×10^3^ N/m^2^)	5.09–392.50	1.97–49.49	0.89–18.94	0.45–4.84

* No specification of the value of hardness is mentioned in the IDDSI framework.

**Table 5 foods-12-03765-t005:** Viscosity range (cP) of liquid foods thickened according to the IDDSI framework compared to NDD and JSDR.

IDDSI Level	Viscosity Ranges (cP)			
	This Study		NDD	JSDR
	Thickened by Thick & EasyTM Clear	Thickened by Corn Starch		
4 (Extremely thick)	645.5–1440.0	929.6–2625.3	>1750	>500
3 (Moderately thick)	244.0–446.7	274.3–483.3	351–1750	300–500
2 (Mildly thick)	95.8–198.1	91.7–153.8	51–350	150–300
1 (Slightly thick)	35.4–79.9	41.6–86.9		50–150
0 (Thin)	1.2–21.2	13.6–14.6	<50	<50

## Data Availability

The data presented in this study are available in Appendix A.

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
