# Peer review of "Quantitative Textural and Rheological Data on Different Levels of Texture-Modified Food and Thickened Liquids Classified Using the International Dysphagia Diet Standardisation Initiative (IDDSI) Guideline"

_foods, 2023, doi:10.3390/foods12203765_

Round 1
Reviewer 1 Report
In general, the results of this manuscript are quite interesting and could benefit the food industry. However, there are some issues that need to be addressed.
Firstly, the display of the unit of viscosity in the manuscript is inappropriate and should be corrected.
Secondly, the conclusion section should be improved. It is recommended that several key findings be added to demonstrate significant research findings.
Additionally, in the last row of Table 4, the hardness values of the tested food items show some overlap, making it challenging to categorize texture-modified food in different schemes that correlate between quantitative and qualitative values. Therefore, there is a need for further explanation and discussion.
In general, the quality of English language is good and easy to understand. However, minor revisions are needed in terms of grammar and typos.
Reviewer 2 Report
In this study, texture-modified food and thickened liquids classified according to the International Dysphagia Diet Standardisation Initiative (IDDSI) guideline were analyzed using a texture analyzer and a rheometer, respectively. The obtained data was analyzed to establish relationships between the IDDSI scale and measurement results. This topic is relatively new and there is not much data in the literature, so this study can help food producers check whether their products can be safely consumed by people with dysphagia. There are some minor corrections that have to be made, mainly related to the data analysis.
Introduction:
IDDSI guideline is mentioned briefly in this section, without going into detail. Most food scientists and engineers are not familiar with this scale, so I suggest authors define the consistency levels from the guideline at this point to make following the text easier.
Material and Methods:
Table S1: It is stated in the table caption that 19 dishes were chosen for TPA in this study; however, only 16 dishes are described in the table. Please correct.
Line 99: Enzyme gellant is not defined. Please state which enzyme was used (e.g. transglutaminase), the name of the producer and the country of origin.
Line 103: Gum-based thickener is not defined. Please state which gum-based thickener was used (e.g. xanthan, guar) – not all producers can purchase the same commercial thickener.
Line 137: “The viscosity of triplicate samples was measured twice.” Does this mean that the authors did six measurements in total? Please explain.
Results:
3.1. Texture profile analysis: Some numbers in Tables S6 and S7 are written in red font, and there is no explanation for it either in the table caption or in the text. Please add the explanation and discuss these results.
Line 153: Does stickiness represent the levels on the IDDSI scale? Please explain.
Line 156: Why is hardness not correlated with stickiness? Those results might be useful. Also, Principal Component Analysis (PCA) might be a more useful tool to explore the correlations between hardness, adhesiveness, cohesiveness and stickiness on the IDDSI scale.
3.2. Rheology analysis: Why did you choose linear regression and exclude levels 0 and 4 from the regression analysis? Maybe the correlation between the flow test and viscosity is polynomial, exponential or logarithmic, please check. Also, please add the obtained equation (correlation between the flow test and the viscosity results) to the text.
Figure 1: Different shapes should also have different colours to better visualize the clustering of the samples.
Figure 2: The results would be easier to interpret if a logarithmic scale (as in Figure 3) is used to represent the data on the y-axis.
Figure 3: With the results presented in Figure 2 and the equation given in the text, this figure is unnecessary.
Discussion:
Table 4: The first row of this table is not understandable, e.g. the last column looks like it is unfinished (“Extensively texture-“). Please correct.
Lines 252-253: Please correct “2” not in subscript for a square meter.
Line 254: “blend-ed” should be “blended”.
Reviewer 3 Report
In this study, quantitative textural and rheological data on different IDDSI 14 levels based on common Chinese ingredients and dishes, which were used to produce 226 samples for the texture analyser and 93 liquid samples for the rheological experiments, were conducted to develop texture-modified food products to promote swallowing safety. The study was considered to be novel in terms of contributing to the classification of product formulations to be developed for a specific consumer group using a mechanistic approach. Different probes and geometries could be used for textural and rheological analysis. Instead of viscosity at 50s conditions, shear dependent viscosity change in the rheometer, i.e. modelling the pseudoplastic character and comparison with the consistency coefficient, could have been preferred. Stability and potential category variability can also be assessed by considering the time factor. However, it is recommended that the study be accepted as it stands, as it will make an important contribution to the literature and help to improve categorization.
Author Response
Thank you very much for taking the time to review this manuscript. We appreciate very much for your comments and support. We will consider your suggestions in future studies.